# OpenReview forum: "MagicMirror: A Large-Scale Dataset and Benchmark for Fine-Grained Artifacts Assessment in Text-to-Image Generation"
_ICLR.cc/2026/Conference — Submitted to ICLR 2026_

### Official Review · Reviewer_Q6jq · 2025-10-21

**Soundness:** 3
**Presentation:** 4
**Contribution:** 4
**Rating:** 6
**Confidence:** 4

**Summary:**

This paper introduces MagicMirror, a large-scale benchmark designed to detect fine-grained artifacts in text-to-image (T2I) generation. The main contribution lies in the creation of MagicData340K, a comprehensive dataset annotated with artifact categories, and the development of MagicAssessor, a vision-language model (VLM) trained to evaluate these artifacts. Finally, the authors present MagicBench, an automated evaluation framework built on MagicAssessor. Experimental results show that their method outperforms existing VLMs in artifact classification and provides valuable insights for future T2I model improvement.

**Strengths:**

* The paper addresses an important but underexplored aspect of T2I evaluation, which goes beyond traditional benchmarks that mainly focus on object presence or semantic alignment.
* The data collection pipeline is well-structured and clearly documented (Figure 4). The use of human annotations, chain-of-thought rationales, and multi-level labeling provides a rich and reliable dataset.
* The experiments are comprehensive, comparing MagicAssessor against both open- and closed-source models. The results demonstrate solid improvements, and the analysis of different artifact categories is informative.

**Weaknesses:**

* While the paper highlights the novelty of the proposed dataset, it would be great to see a clearer comparison with existing benchmarks such as Norma T2I benchmark: GenEval, T2I-CompBench, or Human preference benchmark: Pick-a-Pic, or HPSv2. Some statistics (e.g., dataset size, label diversity, or prompt length) could help readers better understand where MagicData340K stands out.
* The overall accuracy of MagicAssessor, although clearly better than other models, is still on the lower side (some are only ~ 0.3), as shown in Table 2, including element interaction, human and animal anatomy, and object morphology. It would be nice if the authors could discuss how this might affect its use as a reward model in improving T2I systems. Even a brief analysis here would make the potential impact clearer.
* The ablation results and design analysis (especially regarding GRPO and multi-bucket sampling) don’t entirely support the claim that the proposed design is optimal. For instance, in Table 4 (and Table 6 in the appendix), removing multi-bucket sampling improves recall on certain categories like irrational element interaction, while still maintaining competitive scores elsewhere. These patterns suggest that the trade-offs might be more nuanced than described. It would really help if the authors could add more discussion or qualitative examples showing what kinds of issues the proposed design actually solves, which would make the argument much more convincing.

---

**Overall**:
This paper makes a meaningful and timely contribution by introducing a high-quality dataset and benchmark focused on fine-grained artifact detection, which is a topic that has been largely overlooked. Even though the work doesn’t yet show how MagicAssessor directly enhances T2I model training, its methodological thoroughness and dataset quality make it a valuable step forward. Overall, I lean toward accepting the paper.

**Questions:**

The questions are mainly about weaknesses:

* Could authors provide a more explicit quantitative comparison between MagicData340K and existing T2I datasets (e.g., GenEval, T2I-CompBench, Pick-a-Pic, HPSv2)?
* How might the relatively low artifact classification accuracy affect downstream improvements if MagicAssessor were used as a reward model?
* Could authors provide more detailed or qualitative analysis in the ablation section to clarify why the proposed GRPO-based approach works best?

---

> ### Author Response · Authors · 2025-11-21
> **Response to W1: Explicit Comparison with Existing Benchmarks**
>
> We thank the reviewer for requesting a more explicit quantitative comparison. We agree that positioning MagicData340K relative to existing benchmarks is critical for understanding our contribution.
>
> Existing T2I evaluation benchmarks primarily fall into three categories: **Semantic Alignment** (e.g., GenEval, T2I-CompBench), **Human Preference/Aesthetics** (e.g., Pick-a-Pic, HPSv2), and **Coarse Plausibility** (e.g., RichHF). MagicData340K fills a unique gap by focusing on **Fine-Grained Physical and Anatomical Artifacts**.
>
> The comparison highlights our unique advantages:
> 1.  **Scale:** At **340K manually annotated images**, MagicData340K is the largest dataset dedicated to artifact diagnosis, substantially surpassing the scale of the comparable plausibility dataset, RichHF ($\sim 18K$).
> 2.  **Granularity:** Unlike preference benchmarks (Pick-a-Pic), which rely on pairwise rankings, or alignment benchmarks (GenEval), which verify object presence, our dataset offers a **Hierarchical L3 Taxonomy**. This provides an explicit *diagnostic report* (e.g., distinguishing "Hand Structure Deformity" from "Texture Error") rather than a simple pass/fail.
> 3.  **Core Focus:** We shift the evaluation goal from *Does the image follow the instruction?* (Semantic) to *Does the image follow the laws of physics and biology?* (Visual Reality).
>
> The table below summarizes the key differences:
>
> | Benchmark | Focus Area | Label Granularity | Core Label Type |
> | :--- | :--- | :--- | :--- |
> | **MagicMirror (Ours)** | **Physical & Anatomical Artifacts** | **Hierarchical (L3)** | **Diagnosis (e.g., Hand Deformity)** |
> | Pick-a-Pic, HPSv2 | Human Aesthetics/Preference | Pairwise Ranking | Subjective Preference |
> | GenEval | Semantic Alignment | Bounding Box / Binary | Object Presence |
> | T2I-CompBench | Compositional Semantics | Binary / Score | Attribute Binding & Relation |
> | RichHF | Physical Plausibility | Coarse Point | Plausibility Heatmap |
>
> MagicMirror thus provides a necessary foundation for the next stage of T2I research, enabling the community to move from evaluating subjective preference to systematically diagnosing and fixing objective physical flaws.

---

> ### Author Response · Authors · 2025-11-21
> **Response to W2: Impact of Accuracy on Reward Model Utility**
>
> We thank the reviewer for this crucial question regarding the practical application of MagicAssessor. We agree that fine-grained artifact detection is an extremely challenging task, and the current accuracy levels in difficult categories suggest that the reward signal contains noise. However, we argue that MagicAssessor can still serve as a powerful tool for improving T2I systems through targeted application and robust training strategies.
>
> First, the multi-label nature of our model allows for **targeted application**. Developers are not forced to use the **final score**; they can selectively utilize the high-performing categories. For instance, since MagicAssessor excels at detecting *Abnormal Human Anatomy* (as shown in Table 2), it can serve as a reliable "expert reward" for fine-tuning models specifically on portrait generation. This selective usage ensures that the training signal is high-quality where it matters most.
>
> Second, even with noisy signals, **to the best of our knowledge**, MagicAssessor represents the **most comprehensive** automated feedback available for detecting diverse artifact types. In reinforcement learning scenarios like RLHF or DPO, we can mitigate the impact of classification errors by using robust learning techniques. For example, by tuning the KL-divergence penalty or using noise-aware loss functions, we can prevent the generative model from overfitting to specific errors in the reward model while still learning the general direction of improvement.
>
> Finally, beyond its direct role as a reward model, MagicAssessor acts as an **accelerator** for data iteration. The entire framework—including the taxonomy and training pipeline—can be used for "Model-Assisted Labeling." Researchers can use the current model to pre-annotate new data, which human experts then correct. This significantly speeds up the creation of the next generation of larger, higher-precision datasets, creating a positive feedback loop for future improvements. Thus, we see MagicAssessor not just as a final product, but as a foundational tool for the continuous evolution of artifact reduction.

---

> ### Author Response · Authors · 2025-11-21
> **Response to W3: Detailed Analysis of Ablation and Optimality**
>
> We thank the reviewer for the careful examination of our ablation results and for recognizing that the design trade-offs are complex. We agree that the **Total Scores** in Table 4 (and Table 6/7 in the Appendix) can mask the true benefit of the individual components. Our definition of **optimal** for this benchmark task is not based on maximizing a single score but on achieving the **most balanced and robust performance across all artifact categories**, especially mitigating the risk of performance collapse on minority classes.
>
> **1. Multi-Bucket Sampling: Preventing Performance Collapse**
> The reviewer noted that removing Multi-Bucket Sampling might improve *some* metrics. However, this ignores the per-class results (**Table 7 in Appendix**). The true impact is that without this strategy, the model overfits to the majority class (*Abnormal Human Anatomy*), causing the recall on minority classes like *Irrational Element Interaction* to collapse **to near zero** (F1 close to 0.0185). Our proposed sampling strategy is therefore essential for achieving reliable performance across all categories.
>
> **2. Positive Data Resampling: Improving Discrimination**
> This module targets the challenge of **"hard positive" samples** (e.g., anatomically correct but complex hands). Its benefit is a small but critical improvement in both Precision and Recall for *Abnormal Human Anatomy* (Table 7). It forces the model to be more precise by learning to differentiate subtle, correct structures from actual artifacts, preventing it from learning the simple shortcut that "all hands are artifacts."
>
> **3. Consistency Reward: Qualitative Grounding**
> The most convincing evidence for this module is through direct comparison of model outputs. We have added **Figure 9 (Appendix A.6)** to illustrate the issue it solves: "Reward Hacking." This reward forces the model to align its verbal reasoning with its final decision. Without it, the model generates complex, high-scoring rationales that contradict its low-scoring predictions. The Consistency Reward is therefore indispensable for ensuring the model's output is trustworthy and interpretable, which is non-negotiable for an assessment tool.

---

> > ### Comment · Reviewer_Q6jq · 2025-11-21
> > **Response to Authors' Rebuttal**
> >
> > Thank you to the authors for the detailed response.
> >
> > 1. I appreciate the comparison with other datasets and the clear table. It would strengthen the paper even further to also include dataset statistics and place this comparison in the appendix. Since the paper primarily focuses on introducing a new "dataset and benchmark," having a clearer and more complete comparison with existing datasets would make it stand out even more. I believe this can be addressed during the rebuttal period, so I consider this point resolved.
> > 2. I agree with the authors that the proposed model can help address common artifacts. However, being "comprehensive" does not necessarily mean it is suitable for “comprehensive” tasks, i.e., performance and correctness matter just as much. In particular, the reward signal needs to be handled carefully, since it directly affects the loss function, and noisy labels can harm robust training. I also respectfully disagree with the idea that this naturally encourages human–model collaboration. Because certain classes have lower accuracy, their reliability becomes questionable and may require more human verification, which could reduce labeling efficiency. Overall, I think the limitations of the current approach are quite clear, though I agree that the proposed method is more of a tool than a finished product.
> > 3. The explanation provided is reasonable, and I am convinced by the authors’ points.
> >
> > I will keep my original score. From my perspective, a stronger assessor is still needed; however, this paper makes meaningful progress by opening a new discussion and proposing a new benchmark. With that in mind, I lean toward accepting the paper.

---

> > > ### Author Response · Authors · 2025-11-29
> > > **Response to Reviewer Q6jq**
> > >
> > > We thank the reviewer for the constructive feedback and the recommendation to accept our paper.
> > >
> > > 1. Dataset and Benchmark Comparison:
> > > We have added the detailed comparison table in Appendix A.1.4. This table includes statistics like image numbers and prompt sources to clarify how MagicMirror compares with existing datasets and benchmarks.
> > >
> > > 2. Assessor Capability:
> > > We agree that the assessor needs to be stronger. Although MagicAssessor shows high human alignment on the targeted MagicBench, its robustness in complex scenarios can still be improved. We consider this work a starting point for future research on fine-grained artifact detection.
> > >
> > > Thank you again for your support.

---

### Official Review · Reviewer_pHnf · 2025-10-30

**Soundness:** 2
**Presentation:** 2
**Contribution:** 3
**Rating:** 2
**Confidence:** 5

**Summary:**

This paper focuses on evaluating artifacts in text-to-image generation. The authors create a human annotated dataset of generated images containing artifacts and discriminate between different types of artifacts creating a taxonomy. They next propose to fine-tune a model, namely Qwen2.5-VL-7B, via SFT on Chain-of-thought traces extracted from GPT and next GRPO via a multi-reward objective. They show that their fine-tuned model is more accurate in predicting fine-grained artifacts in contrast to off-the-shelf models. Finally, they benchmark different text-to-image generation models on their dataset using their full method.

**Strengths:**

1. The paper presents a human annotated large dataset for artifact evaluation in image generation. This is an important contribution for the community, and a core contribution of the paper is that they also create a taxonomy of artifacts where they target fine-grained evaluation.
2. The authors propose a well thought pipeline for fine-tuning a model on artifact evaluation. Off-the-shelf models do not perform well in recognizing artifacts, as these artifacts might be out of their training distribution. The authors show the improvement that their fine-tuning approach offers on Qwen2.5-VL-7B.

**Weaknesses:**

1. The novelty of the fine-tuning approach for artifact detection, which the authors claim that is one of the main contributions of the paper, is limited. The main novel components of the approach is the data sampling strategy, which is very similar to traditional ML techniques for data imbalances but adapted to the specific task, and the reward combination which is again similar to multi-reward objectives but again adapted to the specific task with specific heuristics derived by the taxonomy of the artifacts.
2. Crucially, the authors do not mention very important details for the creation of their dataset in order to be able to trust the approach and the benchmark. Important details include the inter-annotator agreement and how they are handling disagreements between annotators. Moreover, the paper mentions that one of the artifact categories is Irrational Element Attributes. The authors do not discuss how they discriminate between non factual elements that have been asked in the prompt so they are correctly generated and ones that are not correct. It is very common that prompts include irrational attributes for subjects or objects.
3. The dataset has been created by generating images from specific models, including FLUX.1-dev/schnell, which is one of the best performing models on this benchmark in Table 3. How much does this bias the results? Could it be that images generated by e.g., GPT or Janus are more out of distribution for the fine-tuned evaluator? A human evaluation should be conducted here in order to measure correlation with human ratings and understand the bias of the evaluator.
4. MagicAssessor is trained on a dataset generated by a specific suite of T2I models. Future T2I models may exhibit entirely new or different types of artifacts. How well is the model expected to generalize to these unseen failure modes? Does the "Other Irrationalities" category, which is very small, risk becoming a bucket for many new artifact types, limiting the model's fine-grained utility over time?
5. Not a reason for rejection, but I find it bad that all of related work has been pushed to the Appendix. At least a synopsis should be transferred to the main paper.

**Questions:**

Based on the above weaknesses:
1. Can you explain the details of the dataset for Weakness point 2?
2. Have you analysed in any way the bias as mentioned in Weakness point 3?
3. Can you address the questions of Weakness point 4?

---

> ### Author Response · Authors · 2025-11-21
> **Response to W1: Novelty of the Fine-Tuning Approach**
>
> We thank the reviewer for the feedback on technical novelty. We agree that components like data resampling and hierarchical rewards are adaptations of established strategies. However, we believe our key algorithmic innovation lies in the **Consistency Reward ($r_c$)**, which was designed to solve a specific problem we encountered during training: the "Reasoning-Prediction Mismatch."
>
> When applying GRPO, we found that the model learned to hack the reward system. It would generate detailed but hallucinated reasoning to satisfy length requirements, yet output a safe "Normal" label to maximize accuracy. To address this, we introduced the Consistency Reward to strictly enforce alignment between the chain-of-thought conclusion and the final prediction. In the revised manuscript, we have added **Appendix A.6** and **Figure 9** to visually demonstrate this. The analysis shows that this reward is essential for grounding the model's reasoning in visual evidence.
>
> Beyond specific training techniques, we emphasize that our primary contribution is the **MagicAssessor framework itself**. By effectively combining these strategies, we provide the community with the first scalable, automated tool capable of fine-grained artifact detection, filling a critical gap where general-purpose VLMs currently fail.

---

> > ### Author Response · Authors · 2025-11-21
> > **Response to W5**
> >
> > We thank the reviewer for this constructive suggestion regarding the organization of the paper. In the revised manuscript, we have moved the related work back into the main body.

---

> ### Author Response · Authors · 2025-11-21
> **Response to W2: Dataset Reliability and Artifact Definition**
>
> We thank the reviewer for pointing out these missing details. Establishing trust in the dataset is fundamental to our work. We have updated the manuscript to include the specific protocols and statistics you requested in **Appendix A.1.3** and **Table 6**.
>
> **1. Annotation Agreement and Handling Disagreements**
> To ensure reliability, we did not use a simple single-pass annotation. Instead, we used a rigorous **Iterative Consensus** workflow. Each image was labeled by three independent annotators. We accepted a label only if at least two annotators agreed. For cases where they disagreed (e.g., a 1-1-1 split), we did not simply discard the data. Instead, domain experts analyzed these disputes to clarify the definitions, and the samples were sent back for a second round of labeling.
>
> As shown in the table below (and **Table 6** in the revision), this process yielded high consistency. We achieved **90.43%** agreement in the first round. After the expert review loop, the cumulative agreement rate rose to **97.18%**, confirming the high consistency of our labels.
>
> | Label Category | Round 1 Consensus / Total | Round 1 Rate | Round 2 Cumulative / Total | Round 2 Rate |
> | :--- | :---: | :---: | :---: | :---: |
> | **L1: Normal** | 158,167 / 173,768 | 91.02% | 167,738 / 173,768 | **96.53%** |
> | **L2: Irrational Interaction** | 53,017 / 61,994 | 85.51% | 59,297 / 61,994 | **95.65%** |
> | **L2: Abnormal Human** | 98,741 / 104,289 | 94.68% | 102,537 / 104,289 | **98.32%** |
> | **L2: Abnormal Animal** | 20,941 / 23,464 | 89.52% | 23,009 / 23,464 | **98.06%** |
> | **L2: Abnormal Object** | 30,836 / 37,085 | 83.15% | 35,171 / 37,085 | **94.84%** |
> | **L2: Irrational Attributes** | 706 / 881 | 80.13% | 829 / 881 | **94.10%** |
> | **Total Samples** | **310,418 / 343,269** | **90.43%** | **333,589 / 343,269** | **97.18%** |
>
> *(Note: The "Total Samples" row reflects the unique images confirmed in the dataset.)*
>
> **2. Distinguishing "Artifacts" from "Creative Prompts"**
> We appreciate this insightful observation regarding the distinction between a requested irrationality (e.g., "a flying car") and an unintended artifact. We have carefully addressed this in both our annotation and training phases.
>
> In our annotation guidelines (detailed in **Appendix A.1.1**), we established **Rule 5: Considering User Prompts**. This rule instructs annotators that if an image conforms to special conditions specified in the prompt, it should be labeled as "Normal." For example, if a user explicitly asks for "a cat with wings," that feature is considered correct. If the user asks for a "realistic cat" and it has wings, that is marked as an artifact.
>
> Crucially, our MagicAssessor model is trained to follow this same logic. It takes both the image and the text prompt as input. The model learns to understand the user's intent from the prompt first, ensuring it respects creative requests just as the human annotators did.

---

> ### Author Response · Authors · 2025-11-21
> **Response to W3: Generator Bias and Generalization Analysis**
>
> We deeply appreciate this critical question regarding potential generator bias. To test this, we conducted a **Blind Human Evaluation** comparing "Seen" models (FLUX) against "Unseen" models (GPT-image-1). **The results conclusively refute the bias hypothesis:** MagicAssessor correctly ranks the unseen GPT-image-1 as the top performer (surpassing the seen FLUX) and maintains consistent detection accuracy across both domains (F1 $\approx$ 0.88). This confirms that the model learns universal artifact definitions rather than overfitting to specific generator styles.
>
> To provide specific details, we evaluated 4,000 images generated by 5 representative models using the full MagicBench prompts. The results are reported in **Table 9 (Appendix A.5)** of the revised manuscript.
>
> First, regarding ranking consistency, if the model were biased, it would artificially inflate the scores of the training source (FLUX). However, MagicAssessor correctly ranks the **unseen** `GPT-image-1` (Score: 63.08) as the #1 model, surpassing the **seen** `FLUX.1-dev` (62.16). This ranking aligns perfectly with the ground-truth human evaluation (GPT: 66.37 vs. FLUX: 63.53), achieving an exceptionally high system-level correlation (Pearson $r = 0.99$).
>
> Second, regarding your query about out-of-distribution architectures like Janus, our instance-level analysis confirms robust generalization. As shown in the table below, the F1 score on the unseen GPT-image-1 (0.8809) is virtually identical to the seen FLUX.1-dev (0.8873). Notably, the model achieves its highest accuracy on **Janus-pro** (0.9591), despite its completely different autoregressive architecture. These findings confirm that MagicAssessor detects universal artifact features—such as anatomical logic and physical consistency—regardless of the underlying generator.
>
> | Model | Domain Status | Detection F1 Score |
> | :--- | :---: | :---: |
> | **FLUX.1-dev** | Seen | 0.8873 |
> | **GPT-image-1** | **Unseen** | **0.8809** |
> | **Janus-pro** | **Unseen** | **0.9591** |

---

> ### Author Response · Authors · 2025-11-21
> **Response to W4: Generalization to Future Artifacts and Taxonomy Robustness**
>
> e appreciate this forward-looking question regarding the future relevance of our benchmark. Although text-to-image models are evolving quickly, we argue that our taxonomy is built on fundamental visual rules rather than specific, temporary bugs found in today's models.
>
> To prevent the model from overfitting to current errors, we prioritized maximizing diversity during data construction. We used images from seven different generators, including FLUX, Midjourney, and SD3, to expose the model to the widest possible range of artifacts. As detailed in our response to **W3**, the model successfully generalizes to **Janus-pro**, a model with a completely different architecture. This suggests that MagicAssessor has already learned to recognize general artifact features beyond the specific models it was trained on.
>
> Regarding the concern that the "Other Irrationalities" category might become a catch-all bucket, we view its currently low volume (0.01%) as evidence that our taxonomy is comprehensive. Our main categories—Anatomy, Interaction, and Attributes—are derived from physical and biological laws, which do not change. For instance, if a future model generates a "floating car" where gravity should apply, this would still logically fall under *Irrational Element Interaction* rather than "Other." Therefore, we expect the core structure of the taxonomy to remain stable.
>
> We acknowledge that as models improve, artifacts will likely become more subtle. We envision this evolution as a matter of depth rather than breadth. Our framework serves as a root structure; future iterations can easily accommodate this shift by adding finer sub-labels to the existing categories to capture these detailed errors without needing to overhaul the high-level definitions. Thus, MagicMirror establishes a robust baseline capable of evolving alongside the field.

---

> ### Author Response · Authors · 2025-11-27
> **Follow-up on Rebuttal for Submission 7400**
>
> Dear Reviewer pHnf,
>
> We are writing to follow up on our rebuttal for Submission 7400. We'd like to thank you again for your valuable and constructive comments on our work.
>
> We have submitted our detailed rebuttal and a revised version of the manuscript, in which we have tried our best to address all of your concerns. We were wondering if you might have a chance to look over our response and the revisions.
>
> We look forward to your feedback and are available for any further discussion.
>
> Thank you again for your time and consideration.
>
> Sincerely,
>
> The Authors of Submission 7400

---

### Official Review · Reviewer_LNQe · 2025-11-01

**Soundness:** 2
**Presentation:** 3
**Contribution:** 3
**Rating:** 4
**Confidence:** 4

**Summary:**

This paper proposes a evaluation pipeline for artifacts in text-to-image generation, including human-annotated large-scale dataset of
340K generated images with fine-grained artifact labels, a VLM to assess the generated images, and a benchmark to evaluate current T2I models. It focuses on a critical problem in current text-to-image generation models, and makes great effort to construct a large scale human annotated dataset.

**Strengths:**

1. This paper focuses on the artifacts problem, which is critical and persistent issue in text-to-image generation. This paper provides a whole pipeline with dataset collection, model training, and benchmark construction. The efforts may potentially benefit future works on addressing the artifacts of text-to-image generation.

2. Compared with previous datasets of artifacts, this paper provides more detailed category of artifacts, and propose different level of labeling. This structured categorization provides more precise evaluation signals and enables nuanced analyses of model weaknesses across different artifact types.

3. The proposed dataset is substantial in both size and diversity, with 340k human annotated images, and covers many different types of artifacts in fine-grained levels.

**Weaknesses:**

1. The motivation of describing the artifacts and their location in natural language is unclear. It would be more intuitive to localize artifacts spatially, by marking regions with bounding boxes or segmentation masks. Some artifacts may not be precisely explained merely by natural language, and sometimes their locations may not be easily identified. For example, when the artifacts are only part of a large object, in background region, or texture level distortions.

2. While the dataset provides detailed artifact descriptions, it seems that the framework mainly uses them for binary or categorical classification rather than for region-aware evaluation or feedback. The paper could better leverage these fine-grained signals. For example, to guide targeted model correction or localized reward shaping rather than aggregating them into global scores as in previous paper [1].

3. The distribution of artifact categories in MagicData340K is highly imbalance as shown in Fig 5.

4. The ablation results in Table 4 indicate that removing individual components only slightly affects performance, suggesting that the contribution of each design element is not significant. This weakens the motivation to introduce these strategies.

5. The paper provides limited information about the human annotation process. Key details such as the number of annotators and inter-annotator agreement are missing. Is there a validation process in the annotation, or only one annotator for each sample? Without quantitative measures of consistency, it is difficult to assess the reliability of the large-scale annotations.

[1] Focus-N-Fix: Region-Aware Fine-Tuning for Text-to-Image Generation. (CVPR 2025)

**Questions:**

please see above

---

> ### Author Response · Authors · 2025-11-21
> **Response to W1: Natural Language vs. Spatial Localization**
>
> We sincerely thank the reviewer for this suggestion. **We fully agree that spatial localization (e.g., bounding boxes) is intuitive and valuable; however, we found that bounding boxes are not effective for all artifact cases.** To prioritize the coverage of fine-grained categories and the scale of the dataset, we strategically adopted a classification-based approach for this work, while considering spatial localization as a key direction for future extension.
>
> **1. Limitations of Bounding Boxes for Complex Artifacts**
>
> As the reviewer correctly noted, artifacts often appear in background regions or involve texture distortions. In these scenarios, we found that boxes often fail to capture the *nature* of the error:
> *   **Global & Atmospheric Artifacts:** Issues like *L3: Abnormal Light and Shadow Effect* (e.g., conflicting shadow directions) affect the whole scene. A box covering the entire image conveys no specific information about *what* is wrong. Our label specifically identifies the physical inconsistency.
> *   **Texture & Material Issues:** Consider an image where water looks like solid plastic (*L3: Abnormal Material Texture*). Drawing a box around the water only indicates location. The core issue is a semantic mismatch of material properties, which our taxonomy captures precisely.
> *   **Interaction Failures:** For cases like a hand clipping through a cup (*L3: Abnormal Element Overlap*), a box containing both objects is ambiguous—it is unclear if the hand is wrong, the cup is wrong, or their positioning is wrong. Our classification explicitly defines the *logical error*.
> *   **Scattered Artifacts:** In complex scenes like a crowd, artifacts may manifest as dozens of slightly distorted faces. Drawing boxes for all of them is inefficient and noisy. A unified label like *L2: Abnormal Human Anatomy* is far more robust for evaluating the model's overall capability.
>
> **2. Strategic Trade-off: Precision and Consistency**
>
> To achieve the unprecedented scale of 340K images, we prioritized classification over localization, as drawing precise bounding boxes incurs exponential labor costs and introduces significant ambiguity for abstract artifacts (e.g., defining the spatial boundary of "unnatural lighting"). This strategic choice enabled us to implement a rigorous Iterative Consensus Protocol (detailed in Appendix A.1.3), which achieved a 97.18% consensus rate by the second round, with the remaining hard cases resolved through expert arbitration to ensure dataset-wide consistency. We believe that establishing a universal standard to accurately define "what is wrong" is a foundational and urgent prerequisite to eventually locating "where it is."

---

> ### Author Response · Authors · 2025-11-21
> **Response to W2: Leveraging Fine-Grained Signals for Assessment vs. Repair**
>
> We appreciate the reviewer’s insightful suggestion and the reference to *Focus-N-Fix*. We fully agree that utilizing fine-grained signals for region-aware correction is a valuable direction. However, we would like to clarify that the primary contribution of MagicMirror is to establish a standardized, large-scale **assessment framework** rather than a targeted **correction method**. In this context, our fine-grained classification framework addresses critical evaluation challenges that region-based approaches cannot resolve alone.
>
> First, classification is essential for covering artifact types that are **hard to strictly define with spatial boundaries**. As discussed in our response to **W1**, many critical failure modes—such as *Abnormal Light and Shadow Effect* or *Abnormal Material Texture*—are global or atmospheric in nature. A bounding box cannot effectively capture these issues without ambiguity. By employing a rigorous taxonomy, our framework identifies and quantifies these non-local failures, offering a more comprehensive evaluation perspective than localization methods.
>
> Second, identifying the specific category of an **artifact** provides high-level diagnostic guidance for model development and selection. For developers, our framework reveals whether a model specifically lacks anatomical knowledge, physical interaction logic, or textural consistency. This insight is essential for guiding architectural improvements and data curation strategies. For users, MagicBench enables informed model selection based on specific performance needs, such as prioritizing models with superior scores in *Abnormal Human Anatomy* for portrait tasks.
>
> Finally, MagicAssessor goes beyond simple classification by generating detailed **Chain-of-Thought (CoT)** rationales. This textual reasoning serves as a rich, interpretable signal that explains the nature of the defect. We view our work as the foundational step of establishing a **standardized taxonomy** for artifacts. In future work, we plan to leverage MagicAssessor as a Reward Model to integrate this feedback directly into training loops (e.g., via RLHF), systematically reducing artifact generation.

---

> ### Author Response · Authors · 2025-11-21
> **Response to W3**
>
> We appreciate the reviewer's attention to the data distribution. We acknowledge the class imbalance shown in Figure 5; however, we clarify that this distribution is not a result of sampling bias during dataset construction, but rather a key empirical finding reflecting the real-world behavior of current text-to-image models.
>
> Our prompt collection was rigorously designed to ensure diversity, covering a wide range of subjects from simple objects to complex human scenes. Consequently, the observed frequency of artifact types reveals the actual failure modes of modern generators. For instance, the high prevalence of Abnormal Human Anatomy indicates that generating precise anatomical structures remains the most significant bottleneck in the field. We believe preserving this natural distribution is scientifically valuable for understanding the current state of generative AI.
>
> To ensure that minority classes are still effectively represented, we scaled the dataset to approximately 340,000 images. At this magnitude, even lower-frequency categories contain a sufficient absolute number of samples to support robust statistical analysis and model training. Furthermore, we explicitly addressed the training challenge posed by this imbalance. As described in Section 3.1, we introduced a Multi-Bucket Sampling strategy to balance the learning process. Our ablation study (Table 7 in the Appendix) demonstrates that this strategy effectively prevents the model from overfitting to majority classes, ensuring high recall across all artifact types. Thus, our approach leverages the realistic distribution for benchmarking while mitigating its impact on model training.

---

> ### Author Response · Authors · 2025-11-21
> **Response to W4: Significance of Ablation Components**
>
> We appreciate the reviewer's careful examination of the ablation results. While we acknowledge that the aggregate metrics (Macro Average) in Table 4 might suggest only modest gains, we argue that these top-level numbers mask the critical, specific contributions of each design element. A deeper analysis, particularly of the per-class performance detailed in **Table 7 of the Appendix**, reveals that these strategies are essential for a robust and balanced system.
>
> First, the **Multi-Bucket Sampling** strategy is vital for handling the natural class imbalance of the dataset. Without this strategy, the model tends to overfit to the majority class (*Abnormal Human Anatomy*). As shown in Table 7, removing this component causes the recall for minority classes, such as *Irrational Element Interaction*, to collapse to near zero. The sampling strategy ensures the model remains a versatile evaluator across all categories rather than just a specialist in anatomy.
>
> Second, **Positive Data Resampling** addresses the challenge of "hard positives" (e.g., anatomically correct hands). Without exposing the model to these difficult but correct examples, the model struggles to distinguish between complex structures and actual artifacts. Our results indicate that this strategy improves Recall while maintaining Precision, effectively teaching the model to discriminate rather than simply memorizing that "hands are usually wrong."
>
> Finally, the **Consistent Reward** plays a fundamental role in the stability of GRPO training. Beyond the quantitative improvements in F1 score, its most significant contribution is qualitative. As illustrated in the newly added **Figure 9 (Appendix A.6)**, removing this reward leads to "reward hacking," where the model generates hallucinated reasoning to satisfy length requirements while outputting a safe "Normal" label. The Consistent Reward is therefore indispensable for ensuring that the model's rationales are trustworthy and logically aligned with its predictions.

---

> ### Author Response · Authors · 2025-11-21
> **Response to W5: Annotation Process, Validation, and Consistency**
>
> We apologize that the specific details regarding our quality control were not sufficiently covered in the original text. We agree that quantitative measures are essential for assessing reliability. To address this, we have added **Appendix A.1.3** and **Table 6** in the revised manuscript to provide the granular details and statistics.
>
> **1. Number of Annotators and Validation Process**
>
> To answer your specific questions: **No, we did not rely on a single annotator per sample.** Instead, we employed a rigorous **Iterative Consensus** workflow designed to actively resolve subjectivity. The process consisted of the following stages:
> *   **Parallel Labeling (3 Annotators):** Each image was independently labeled by **three different annotators** simultaneously.
> *   **Validation via Majority Voting:** A sample was accepted into the dataset only if **$\ge 2$ annotators agreed** on the label.
> *   **Batch-Level QA:** Completed batches underwent random inspection by experts. If a batch failed, it was rejected entirely and re-labeled.
> *   **Expert Refinement Loop:** Crucially, samples that failed to reach consensus (e.g., 1-1-1 splits) were not simply discarded. They were analyzed by domain experts to refine the taxonomy definitions (e.g., clarifying boundary cases) and then re-evaluated in a second round.
>
> **2. Quantitative Consistency Measures**
>
> As shown in the table below (and **Table 6** in the revision), this rigorous process yielded high reliability. Attributable to strict pre-training, **90.43%** of samples reached majority consensus immediately in Round 1. Subsequently, after the expert refinement loop in Round 2, the **Cumulative Consensus Rate rose to 97.18%**. This high convergence rate demonstrates that our large-scale annotations are robust and reliable.
>
> | Label Category | Round 1 Consensus / Total | Round 1 Rate | Round 2 Cumulative / Total | Round 2 Rate |
> | :--- | :---: | :---: | :---: | :---: |
> | **L1: Normal** | 158,167 / 173,768 | 91.02% | 167,738 / 173,768 | **96.53%** |
> | **L2: Irrational Interaction** | 53,017 / 61,994 | 85.51% | 59,297 / 61,994 | **95.65%** |
> | **L2: Abnormal Human** | 98,741 / 104,289 | 94.68% | 102,537 / 104,289 | **98.32%** |
> | **L2: Abnormal Animal** | 20,941 / 23,464 | 89.52% | 23,009 / 23,464 | **98.06%** |
> | **L2: Abnormal Object** | 30,836 / 37,085 | 83.15% | 35,171 / 37,085 | **94.84%** |
> | **L2: Irrational Attributes** | 706 / 881 | 80.13% | 829 / 881 | **94.10%** |
> | **Total Samples** | **310,418 / 343,269** | **90.43%** | **333,589 / 343,269** | **97.18%** |
>
> *(Note: The "Total Samples" row reflects the unique images confirmed in the dataset.)*

---

> ### Author Response · Authors · 2025-11-27
> **Follow-up on Rebuttal for Submission 7400**
>
> Dear Reviewer LNQe,
>
> We are writing to follow up on our rebuttal for Submission 7400. We'd like to thank you again for your valuable and constructive comments on our work.
>
> We have submitted our detailed rebuttal and a revised version of the manuscript, in which we have tried our best to address all of your concerns. We were wondering if you might have a chance to look over our response and the revisions.
>
> We look forward to your feedback and are available for any further discussion.
>
> Thank you again for your time and consideration.
>
> Sincerely,
>
> The Authors of Submission 7400

---

### Official Review · Reviewer_krp7 · 2025-11-01

**Soundness:** 4
**Presentation:** 4
**Contribution:** 3
**Rating:** 6
**Confidence:** 4

**Summary:**

The paper introduces a complete stack for artifact assessment in T2I: (1) a fine-grained taxonomy of artifacts and a 340K human-annotated dataset called MagicData340K, (2) a specialized assessor named MagicAssessor based on Qwen2.5-VL-7B fine-tuned using SFT followed by GRPO with custom sampling and rewards, which produces labels and rationales, and (3) an automated benchmark named MagicBench that applies MagicAssessor to evaluate a set of T2I models across human, animal, object, and interaction categories. Results show that MagicAssessor clearly outperforms general-purpose VLMs in artifact recognition, and MagicBench indicates that even leading generators such as GPT-image-1 still produce significant artifact rates.

**Strengths:**

S1: The fine-grained and hierarchical taxonomy with large-scale human annotation is impressive. The L1 to L3 scheme, moving from normal versus artifact to anatomy, attributes, and interaction, then to hand-level and similar details, along with curated guidelines, represents a clear improvement over coarse plausibility dots.

S2: Looks like the dataset is of high quality.

S3: The design of the artifect-based reward looks good to me.

S4: The paper looks really nice, very well-written.

**Weaknesses:**

W1: The paper describes expert guidelines and oversight but does not report inter-annotator agreement such as kappa or Fleiss scores, nor re-label consistency metrics, despite the subjective nature of fine-grained labels like element overlap versus low-quality area. I recommend the authors include this in the rebuttal.

W2: A subset of rationales is synthesized by GPT-4o from human descriptions to bootstrap chain-of-thought. While pragmatic, the paper does not specify the fraction of CoT data or analyze how GPT-4o phrasing may bias the learned reasoning style. If the evaluator later favors assessor-like wording, this may create a stylistic loop. It is important to examine the CoT steps. If possible, the authors should add this point during rebuttal.

W3: ​​The consistency reward is meant to reduce reward hacking, but the evidence is mainly aggregate F1 changes. There is no audit of failure cases such as verbose rationales that still mislabel. GRPO can overfit to reward signals shaped by noisy heuristics like formatting or high-level label focus. No analysis on this is given in the paper.

W4: MagicData340K includes images from several generators such as FLUX, Kolors, SD3 and SD3.5, Midjourney, and internal sources, but it is unclear whether generator identity leaks into labels, for example via artifacts that appear recognizably FLUX-like. This may let MagicAssessor rely on style-based priors. More broadly, the test set of 17366 samples is not broken down clearly in terms of size relative to training split or coverage across generator and style diversity.

W5: Although the paper overall looks very nice, figure 2 does not match the style of the other figures. It appears too simple and a bit unattractive.

**Questions:**

See the previous weakness.

Could the authors report inter-annotator agreement (e.g., Cohen’s κ or Fleiss’ κ) for a subset of the dataset?

What proportion of CoT rationales are human-authored vs. GPT-4o-synthesized, and how do their linguistic or reasoning styles differ? I guess some of CoT should be checked instead of completely relying on the automatically generated.

How many unique T2I generators are represented in the dataset, and is there any overlap between generators used for training and those used in MagicBench?

Did the authors observe cases where GRPO-trained models produced plausible but factually incorrect rationales to maximize rewards?

---

> ### Author Response · Authors · 2025-11-21
> **Response to W1: Inter-Annotator Agreement and Label Consistency**
>
> We thank the reviewer for highlighting the critical need for quantitative reliability metrics. We fully agree that for fine-grained tasks with inherent subjectivity, a standard one-pass annotation is insufficient. Sharing this concern, we originally designed our data collection with a rigorous "Iterative Consensus Protocol" to ensure high reliability. We have added Appendix A.1.3 and Table 6 in the revised manuscript to provide the missing details of this process.
>
>
> **1. Pipeline Clarification (Reference to Figure 4)**
>
> We apologize that the specific protocols of our annotation quality control were not sufficiently detailed in the original text. We clarify that our rigorous "Iterative Consensus" pipeline corresponds to the Annotation Training and Sampling Inspection modules visualized in the middle section of Figure 4. To address this, we have added Appendix A.1.3 and Table 6 in the revised manuscript to provide the granular protocols and statistics for this workflow.
>
>
> **2. Why "Iterative Consensus" instead of Static Kappa?**
>
> Instead of a single-pass annotation (where Cohen’s $\kappa$ is typically calculated), we employed a multi-round, expert-in-the-loop workflow to actively resolve subjectivity. Our goal was not just to *measure* disagreement, but to *eliminate* it through rule refinement. The process involved four key stages:
> *   **Stage 1: Rule Iteration & Pilot Alignment:** Annotators participated in a unified pilot phase aligned with the formal distribution to ensure rule comprehension before the main launch.
> *   **Stage 2: Parallel Labeling with Majority Voting:** Each sample was annotated by three independent annotators. A sample was accepted only if **$\ge 2$ annotators agreed** (Majority Consensus).
> *   **Stage 3: Batch-Level Expert QA:** Batches failing QA were rejected entirely and re-labeled.
> *   **Stage 4: The Refinement Loop:** Crucially, samples failing consensus (e.g., 1-1-1 splits) were not discarded. We treated these as critical **boundary cases**. Experts analyzed them to refine the taxonomy (e.g., clarifying *Abnormal Anatomy* vs. *Blurry Texture*), and these samples underwent a **second round** of labeling.
>
> **3. Quantitative Consensus Statistics**
>
> As shown in the table below (and Table 6 in the revision), this rigorous process yielded high reliability. Attributable to the strict pre-training and pilot alignment described in Stage 1, 90.43% of samples reached majority consensus immediately in Round 1. Subsequently, after the expert refinement loop in Round 2, the Cumulative Consensus Rate rose to 97.18%. This high convergence rate demonstrates the reliability of our labels, validated by both majority vote and expert arbitration.
>
>
> | Label Category | Round 1 Consensus / Total | Round 1 Rate | Round 2 Cumulative / Total | Round 2 Rate |
> | :--- | :---: | :---: | :---: | :---: |
> | **L1: Normal** | 158,167 / 173,768 | 91.02% | 167,738 / 173,768 | **96.53%** |
> | **L2: Irrational Interaction** | 53,017 / 61,994 | 85.51% | 59,297 / 61,994 | **95.65%** |
> | **L2: Abnormal Human** | 98,741 / 104,289 | 94.68% | 102,537 / 104,289 | **98.32%** |
> | **L2: Abnormal Animal** | 20,941 / 23,464 | 89.52% | 23,009 / 23,464 | **98.06%** |
> | **L2: Abnormal Object** | 30,836 / 37,085 | 83.15% | 35,171 / 37,085 | **94.84%** |
> | **L2: Irrational Attributes** | 706 / 881 | 80.13% | 829 / 881 | **94.10%** |
> | **Total Samples** | **310,418 / 343,269** | **90.43%** | **333,589 / 343,269** | **97.18%** |
>
> *(Note: The "Total Samples" row reflects the unique images confirmed in the dataset.)*

---

> > ### Author Response · Authors · 2025-11-21
> > **Response to W2: CoT Data Composition and "Stylistic Loop" Concerns**
> >
> > We thank the reviewer for this insightful question regarding the risk of a "stylistic loop." We address this by clarifying our data construction protocol and training strategy.
> >
> > **1. Protocol: 100% Human-Guided Hybrid Synthesis**
> >
> > To answer the specific question on proportion: **100% of the 1,294 CoT samples are hybrid**, combining human observations with AI formatting.
> > *   **Human Grounding:** The core reasoning content (the *what* and *where* of the artifact) was explicitly provided by human experts.
> > *   **Style Control via ICL:** GPT-4o served only to structure these facts. **Regarding linguistic differences:** Raw human descriptions are often fragmented (e.g., "extra finger on left hand"). GPT-4o, constrained by **In-Context Learning (ICL)** with expert-authored exemplars, unified these into a coherent, structured narrative. Thus, the reasoning style reflects our designed standard, not GPT's default bias.
> > *   **Quality Check:** A final manual review ensured the output remained faithful to the human input. Due to the strict control in the preceding steps, only a very small number of samples were rejected at this stage, which is why this specific verification step was not emphasized in the original manuscript.
> >
> > **2. Mitigating Bias via Training Strategy**
> >
> > We designed a two-stage training pipeline to explicitly prevent stylistic overfitting:
> > *   **Minimal SFT for Format Adaptation:** The SFT phase was intentionally kept "light" with minimal training steps. Its primary role was a **"Cold Start"** to teach the model the output format (syntax), rather than embedding deep reasoning behaviors.
> > *   **Breaking the Loop via GRPO:** The "stylistic loop" is actively broken in the subsequent GRPO stage. Since the optimization is driven by an **outcome-based reward** (the correctness of the final label) rather than textual similarity, the model is forced to ground its reasoning in actual visual evidence to maximize rewards, rather than merely mimicking GPT-style phrasing.
> >
> > By combining human-grounded inputs with outcome-based reinforcement learning, we ensure the model learns *how to reason*, not just *how to sound like* GPT-4o.

---

> ### Author Response · Authors · 2025-11-21
> **Response to W4: Generator Bias, Data Composition, and Generalization**
>
> We thank the reviewer for raising this critical question regarding potential generator bias. It is vital to ensure our model learns to find artifacts rather than just memorizing specific model styles. We address this with dataset details and **new experimental evidence**.
>
> **1. Dataset Composition & Strict Separation**
>
> To answer your specific questions:
> *   **7 Unique Generators:** MagicData340K consists of images from 7 distinct sources: FLUX.1-dev/schnell, Kolors, SD3, SD3.5, Midjourney, and an internal model.
> *   **No Data Leakage:** We ensure strict separation between training and testing. The test set (17,366 samples) contains **zero sample overlap** with the training set.
> *   **Model Overlap:** While MagicBench includes models seen during training (to test retention), we explicitly added **Unseen Models** (GPT-image-1, Seedream3.0, Janus-pro) to rigorously test generalization.
>
> **2. Proof of Generalization: Human vs. Model Correlation**
>
> To definitively test whether MagicAssessor relies on "style priors" (e.g., unfairly favoring seen FLUX images), we conducted a **Blind Human Evaluation**. The results are detailed in **Table 9 of Appendix A.5** in the revised manuscript.
>
> *   **Ranking Consistency (No Bias towards "Seen" Models):**
>     MagicAssessor does not blindly favor the models it was trained on. It correctly identifies that the **unseen** model `GPT-image-1` (Score: 63.08) is better than the **seen** model `FLUX.1-dev` (62.16). This matches the human expert judgment perfectly (GPT: 66.37 vs. FLUX: 63.53).
>
> *   **Consistent Accuracy on Unseen Architectures:**
>     We compared the detection accuracy (F1 Score) on seen vs. unseen models. As shown in the table below, the performance gap is insignificant. Notably, the model performs exceptionally well on **Janus-pro**, which uses a completely different architecture (autoregressive) from the diffusion models in the training set.
>
> | Metric | FLUX (Seen) | GPT-image-1 (**Unseen**) | Janus-pro (**Unseen**) |
> | :--- | :---: | :---: | :---: |
> | **Overall F1 Score** | 0.887 | **0.881** | **0.959** |
>
> The perfect alignment with human rankings and the high accuracy on unseen models prove that MagicAssessor detects **universal artifact patterns** (like anatomical errors) rather than relying on specific generator styles.

---

> ### Author Response · Authors · 2025-11-21
> **Response to W3: Reward Hacking and Consistency Reward**
>
> We appreciate the reviewer's deep insight into the dynamics of RL training. To address your concern about the lack of failure case audits, we have added **Appendix A.6** and **Figure 9** in the revised manuscript to provide the requested qualitative analysis.
>
> We explicitly confirm your hypothesis: without the Consistency Reward, the model indeed learned to **hack the reward system**. Specifically, we observed a "Reasoning-Prediction Mismatch" where the model decoupled its reasoning from its conclusion. It would generate a detailed, critical analysis finding specific artifacts (satisfying the CoT length requirements), but then output "Normal" in the final label (the majority class) to maximize the prediction accuracy reward. This allowed the model to satisfy the format requirements while defaulting to the safest prediction, effectively bypassing the harder task of consistent diagnosis.
>
> The Consistency Reward ($r_c$) was designed precisely to penalize this mismatch. By verifying that the *conclusion of the reasoning* matches the *final label*, $r_c$ prevents this strategy. As shown in the bottom row of **Figure 9**, adding $r_c$ forces the model to align its final judgment with its visual analysis.
>
> Regarding your concern about shortcuts, we argue that our system constraints prevent this overfitting. **Following the format strictly** is a necessity for parsing, not a noise signal. Furthermore, our reward system reflects the **natural taxonomy of our dataset**, where L3 labels are specific sub-types of L2 categories. The model cannot simply guess a high-level category (L2) to score well; it is rewarded mainly when the fine-grained details (L3) are also correct. This forces the model to learn from bottom-up details rather than relying on top-down priors.

---

> ### Author Response · Authors · 2025-11-21
> **Response to W5: Aesthetic Improvement of Figure 2**
>
> We thank the reviewer for the feedback on our figure design. We agree that the original Figure 2 was too simple and did not match the visual quality of the rest of the paper.
>
> We have redesigned it to appear more professional and stylistically consistent. Please refer to the updated **Figure 2 on Page 4** of the revised manuscript.

---

### Author Response · Authors · 2025-11-29
**Summary of Rebuttal and Revisions**

We summarize our responses and revisions as follows:

### **1. Core Issues**

**1) Annotation Consistency (`krp7.R1.W1`, `LNQe.R2.W5`, `pHnf.R3.W2`)**

To ensure consistency, we implemented strict pre-annotation training, along with rigorous filtering and multi-round iteration during annotation. Specifically, inconsistent data were collected, rules were refined, and samples were re-labeled until all samples were completed, **to avoid discarding "hard cases" that are difficult to annotate**. We have added the detailed process, rules, and statistics in **Appendix A.1.3** and **Table 6** to supplement the initial manuscript.

**2) Comparison with Human Evaluation (`krp7.R1.W4`, `pHnf.R3.W3`)**

We added human evaluation experiments. The results demonstrate that our model achieves **high consistency with human evaluation** and performs well on unseen data, proving its generalization within and outside the training domain. Relevant content has been added to **Appendix A.5** and **Table 10**.

**3) Effectiveness of Module Design (`krp7.R1.W3`, `LNQe.R2.W4`, `Q6jq.R4.W3`)**

The module design does not aim solely to improve the L1 level F1 Score. **Multi-Bucket Sampling** achieves better balance, avoiding F1 scores close to 0 for certain classes (see **Table 8**). The **Consistency Reward** improves the F1 Score while providing more confident Rationales. We have added discussion and examples in **Appendix A.6** and **Figure 9**.

### **2. Task Definition**

**1) Why Classification Instead of Bounding Boxes? (`LNQe.R2.W1`, `LNQe.R2.W2`)**

Previous work like RichHF-18K used heatmaps and overall scores. Our work focuses on artifacts and provides a fine-grained, multi-level classification. While bounding boxes are meaningful for some types, they are not applicable to all artifact types. To ensure **uniformity and scalability**, we prioritized fine-grained categorization, forming **MagicData340K**.

**2) Is Text-Image Inconsistency Considered an Artifact? (`pHnf.R3.W2`)**

No. As stated in Rule 5 of the annotation guidelines in the original **Appendix A.2** (now **Appendix A.1.1**), user prompts must be considered. During MagicAssessor training, the T2I prompt is also included in the input.

**3) Generalization to Future Artifacts ("Other Irrationalities"): (`pHnf.R3.W4`)**

Our definition of artifacts is based on a **taxonomic perspective**, not merely induced from a limited dataset. We agree that as T2I models evolve, more detailed artifact categories will need to be considered. However, we believe these finer categories are more likely to emerge as refinements under the existing **L3 labels**, rather than falling into the very small "Other Irrationalities" category (0.01%).

### **3. Other Discussions**

**1) Effect of Single Style in CoT Cold Start Data Construction(`krp7.R1.W4`)**

During synthesis of CoT rationales, we provided real annotation details and output examples (in human handwriting style) and we examed all the final CoT rationales. We used this data only for **cold start** to teach the model to output right-form reasoning; the training steps were minimal. The reward targets the **result**, not the style. The model focuses on the consistency between the reasoning process and the result, as well as the correctness of the result, leaving the style flexible.

**2) Class Imbalance in Dataset (`LNQe.R2.W3`)**

Our dataset is derived from widely covered prompts and multiple T2I models, so the distribution reflects the **real-world situation**. We annotated a large scale to ensure that even minority categories have a sufficient amount of data.

**3) Novelty of Module Design (`pHnf.R3.W1`)**

How to apply GRPO to this specific classification task is underexplored. All three modules were proposed to address practical issues encountered during experiments. We agree that Multi-Bucket Sampling and Positive Data Resampling apply existing methods to our task, but the **Consistency Reward** was independently proposed to address sparse rewards in GRPO. We understand that calling them "innovations" might be considered inappropriate by the reviewer, so we have corrected this to "**components**" in the revised manuscript.

**4) Low Accuracy on Some Categories (`Q6jq.R4.W2`)**

We agree that MagicAssessor's discrimination capability in complex scenarios needs improvement. However, it is worth noting that MagicBench results show **high consistency with human evaluation**, indicating that the discrimination capability is functional in targeted scenarios.

### **4. Revisions on Presentation**

**1) Figure 2 Quality (`krp7.R1.W5`):** We have redrawn Figure 2 to be more visually appealing.

**2) Related Work Location (`pHnf.R3.W5`):** We have moved the Related Work section back to the main text.

**3) Comparison with Other Datasets (`Q6jq.R4.W1`):** Added in **Appendix A.1.4** and **Table 7** of the latest manuscript.

---

### Meta-Review · Area_Chair_yyS3 · 2026-01-14

**Summary:**

This paper introduces MagicMirror, a framework for artifacts accessment in text-to-image generation tasks. The major concerns from the reviewers include: 1. Effectiveness of modules. 2. Alignment with human evaluations. 3. Motivation of describing the artifacts and their location in natural language. 4. Imbalance of data distribution.

The rebuttal addresses a portion of the concerns. However, some major concerns are not resolved convincingly. Therefore, I suggest to reject this paper.

**Reviewer Concerns:**

The concerns on the effectiveness of modules and the alignment with human evaluations have been addressed in the rebuttal. However, the responses to the remaining important concerns such as the motivation of describing the artifacts and their location in natural language and the imbalance of data distribution are not convincing enough.

**Reviewer Scores:**

Few reviewers participated in the discussion after the rebuttal, but did not change the score.

---

### Decision · Program_Chairs · 2026-01-26

Reject